# Advancements in Thyroidectomy: A Mini Review

Woochul Kim [1] , Ja Kyung Lee [1], Hyeong Won Yu [2] and June Young Choi [2,*]

1    Department of Surgery, Seoul National University Bundang Hospital, 82, Gumi-ro 173 Beon-gil, Bundang-gu, Seongnam-si 13620, Gyeonggi-do, Republic of Korea
2    Department of Surgery, Seoul National University Bundang Hospital and College of Medicine, 82, Gumi-ro 173 Beon-gil, Bundang-gu, Seongnam-si 13620, Gyeonggi-do, Republic of Korea
*    Correspondence: juneychoi@snubh.org; Tel.: +82-31-787-7107

**Abstract:** Demand for minimally invasive surgery has driven the development of new gadgets and surgical techniques. Yet, questions about safety and skeptical views on new technology have prevented proliferation of new modes of surgery. This skepticism is perhaps due to unfamiliarity of new fields. Likewise, there are currently various remote-access techniques available for thyroid surgeons that only few regions in the world have adapted. This review will explore the history of minimally invasive techniques in thyroid surgery and introduce new technology to be implemented.

**Keywords:** thyroid; robot; minimally invasive surgical procedures; artificial intelligence

## 1. Introduction

According to the latest global cancer statistics conducted in 2018, thyroid cancer is responsible for 567,000 cases annually and ranks 9th frequently diagnosed cancer worldwide [1]. Today, the survival rate of thyroid cancer sits above most other cancers [2,3] and patients enjoy various treatment options. However, this was not the case when thyroidectomy was first introduced. It was considered one of the toughest organs to extract as surgeons had to face massive bleeding and infection which led to high mortality rates. It was not until in 19th century when developments of artery forceps, antisepsis, and ether has allowed safer dissection [4,5]. Then, surgeons diverted from radical dissection to reduce post-operative complications such as vocal cord paralysis and hypoparathyroidism. The paradigm has gradually shifted to maintaining patients' quality of life after surgery. These changes, added with patients' demand for more aesthetic outcomes, have driven development of remote approaches to thyroidectomy we see today.

## 2. From Endoscopic Approaches to Robotic Surgery

Developing a novel, yet technically feasible surgical approach requires advances in various scientific fields. A fiberoptic camera have had to been developed before laparoscopic surgery is attempted. Moreover, surgeons would have had envisioned how these new instruments would work in a pre-designed surgical space. The early pioneers of robotic thyroidectomy have faced various obstacles from choosing the most adequate incision sites to hide and minimize visible scars, to approaching the thyroid from distantly located port sites. Currently, there are three most used remote approaches to robotic thyroidectomy. In order of development, they are trans-axillary approach, bi-axillo-breast approach (BABA), and the natural orifice surgery.

### 2.1. Trans-Axillary Approach

Trans-axillary thyroidectomy was first described by Ikeda et al., in 2001 [6]. The surgery was performed by inserting a 12 mm trocar through a 3 cm incision below the inferior border of the clavicle, and two 5 mm trocars by each side (Figure 1). Their initial findings showed comparable results to open thyroidectomy regarding the amount of blood

loss, or the length of hospital stay. This study has proven feasibility and safe application of trans-axillary approach to thyroidectomy. Later a gasless approach was developed by Yoon et al., where skin flap dissection is first made to secure a working space before the actual thyroidectomy. This skin flap is maintained by an external lifting device to create an open space for instrument access [7]. With larger skin flap than the gas inflated approach, the gasless approach is known to have better visualization of the surgical field. Moreover, it can avoid $CO_2$-related complications such as respiratory acidosis, hypercapnia, and subcutaneous emphysema. A randomized controlled trial evaluated operation time, estimated blood loss, complication rates and cost effectiveness between endoscopic trans-axillary thyroidectomy and open thyroidectomy. Their findings showed no significant differences between the two groups except for cost and operation time [8]. The same results were found in the systemic meta-analysis [9]. Due to additional steps for making a skin flap, endoscopic surgery was found to take additional operation time and cost compared to open surgeries. Regarding oncologic outcomes for endoscopic trans-axillary approach, a 10 year follow-up of study showed that their oncologic outcomes were also compatible with the conventional open thyroidectomy [10]. However, this study only included micropapillary thyroid carcinomas without lateral neck metastasis. Moreover, the long-term oncologic outcomes of modified radical neck dissection using the endoscopic trans-axillary approach is yet to be evaluated.

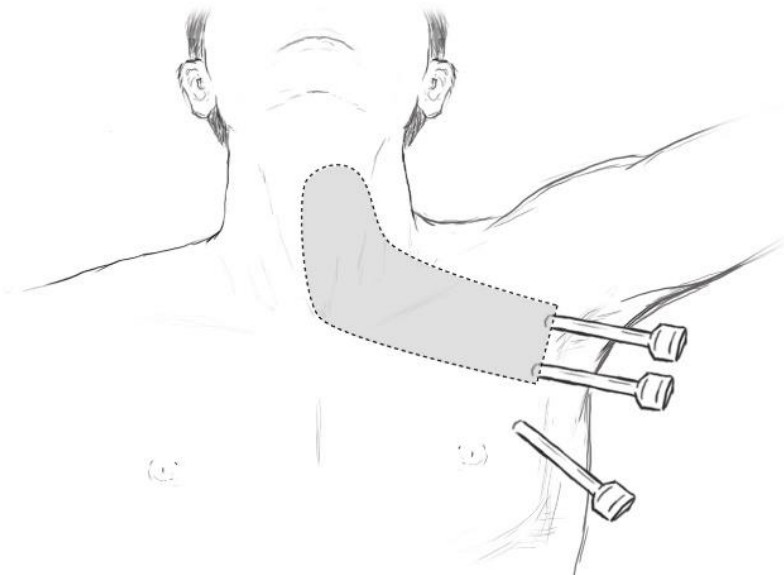

**Figure 1.** Port insertion sites relative to anterior neck.

The first robotic trans-axillary thyroidectomy was performed in 2007 by Kang et al., in Korea and their first 100 patient experience was published in 2009 [11]. They adapted the robotic console by simply docking the arms to where the endoscopic ports have been inserted. The reason for this adaptation was to overcome limitations of endoscopic surgery by improving dexterity, ergonomics, and visualization [12–14]. The adaptation of the robot has proven useful when articulation of the robotic arms has enabled surgeons to perform single axillary access total thyroidectomy [15,16] and lateral neck dissection [17,18]. Kim et al., then further modified robotic approach and reported their initial 200 cases of single port trans-axillary robotic thyroidectomy in 2022 [19]. Though robotic trans-axillary thyroidectomy has proven to be technically feasible over the years, its efficacy is yet to be proven. The robotic approach has shown no significant differences in surgical outcomes to the endoscopic approach [14,20]. Rather, some studies have found higher rates of hypocalcemia, and longer duration of paresthesia from the robotic approach compared to open thyroidectomy. Moreover, since robotic surgery is still performed mainly on patients that do not show extensive lateral neck metastasis, its oncologic safety is still under debate.

### 2.2. Bilateral Axillo-Breast Approach (BABA)

The BABA approach is the modified version of the axillo-bilateral breast approach (ABBA) developed by Shimazu et al. [21,22]. This new method was first introduced by Choe et al., in 2007 in Korea where it utilized two 5 mm and two 12 mm ports through both axillae and areolar incisions (Figure 2). It differs from the trans-axillary approach in that both thyroid lobes are well visualized similar to open thyroidectomy. Therefore, it is thought to provide easier access to both thyroid lobes without inserting additional ports. The evaluation of first 110 cases of the BABA showed that there were in total 10 post-operative complications recorded; most complications were transient hypocalcemia, with one case of pneumothorax. Later paper by Chung et al., compared the outcomes of BABA thyroidectomy with open thyroidectomy in 2007 [23], which showed no statistical differences in post-operative complications except for higher numbers of transient recurrent laryngeal nerve palsy. Further study evaluating endoscopic BABA surgery published by Kim et al., in 2010 on completion thyroidectomy proved safety and feasibility of the BABA approach [24]. Yet, those studies mostly included patients without lateral neck metastasis or large thyroid tumors (>4 cm). Therefore, the feasibility of endoscopic BABA operation for extensive lateral neck dissection is still unproven.

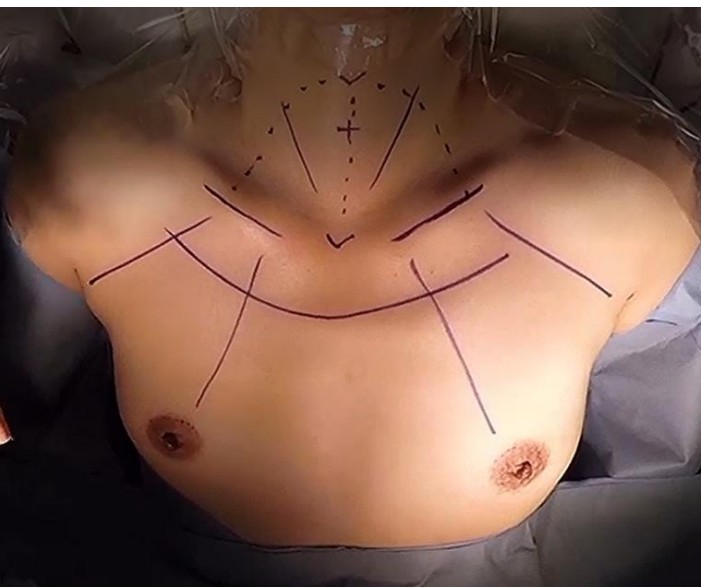

**Figure 2.** Port insertion sites of endoscopic bilateral axillo-breast approach (BABA).

The initial experience of robotic thyroidectomy using the da Vinci System using the BABA technique was published in 2009 by Lee et al. [25]. Just like the first robotic approach to trans-axillary thyroidectomy, the robotic BABA surgery simply docked robotics arms to the already existing 4 ports. Immediate improvements from adapting robotic console were seen during the superior pole dissection, where articulation of the robotic arm allowed safer coagulation of arteries and veins compared to the rigid endoscopic arms. This articulation also proved effective for approaching deep spaces, which helped perform lateral neck dissections [26–28]. Their immediate post-operative outcomes proved to show less post-operative pain and improved cosmetic outcomes and compared to open thyroidectomy. It also showed non-inferior results when comparing surgical complications such as hypoparathyroidism and recurrent laryngeal nerve palsy. [29,30]. When comparing the robotic BABA surgery to endoscopic BABA surgery, a propensity score analysis showed less operation time, but lower rates of recurrent laryngeal nerve palsy and lower stimulated Tg levels after thyroidectomy [31,32]. On the other hand, utilizing a robotic console still comes with higher costs with longer operation time and requires careful patient selection than open thyroidectomies [29,30,33]. Therefore, it is still questionable whether these disadvantages are worth having hidden scars, when a recent study by Chen et al., showed

patients' perception of scars do not differ from having a minimal scar on the anterior neck [34].

### 2.3. Natural Orifice Surgery

As minimally invasive surgery gained interests by patients and surgeons alike, more people adapted the axillary approach and the breast areolar approaches to thyroidectomy. However, some surgeons have questioned their minimally invasiveness due to their extensive amount of skin flap created to perform thyroidectomy [34]. So, in 2009, Benhidjeb and Wilhem et al., introduced first ever endoscopic transoral thyroidectomy [35,36]. Before operation, patients were intubated naso-tracheally and were given pre-operative antibiotics before operation. Surgery itself utilized three ports inserted into three incisions at bilateral vestibules of the mouth and a single incision sublingually to patients with benign thyroid nodules. However, this approach saw technical difficulties with alarming rates of complications [37–39]. Anuwong saw limitations to this approach seeing that ports pierced through the floor of the mouth is what was causing massive tissue damage. He then proposed a vestibular approach (TOETVA) in 2016, dramatically reducing complications [40,41]. He later published initial outcomes of 400 cases of TOETVA in 2018, which showed similar surgical outcomes and complication rates compared to open thyroidectomy [42].

As with other approaches, the transoral approach has adapted robotic console to enhance surgical maneuvering. The first application of robotic surgery for transoral thyroidectomy was established by Richmon et al., in 2010 on a cadaver model [43]. Later, a case series of robotic transoral thyroidectomy on live patients was published in 2011 [44]. The authors found that the sublingual incision limited camera rotation and saw high complication rates. Nakajo et al., was one of the first surgeons to explore the vestibule access to transoral thyroidectomy, where they found significantly decreased complication rates with better surgical movement [45] (Figure 3). As TOETVA provided safer access to thyroid glands through the transoral cavity, more surgeons are adapting robotic console to TOETVA. Robot has allowed for meticulous movement with enhanced access to hard-to-reach spaces. This has allowed for the transoral approach to lateral neck dissection [46]. However, transoral surgeries for cancers are still limited to highly selected patients [47,48], with limited data on long-term outcomes. Moreover, postoperative seroma, infection, and relatively small working spaces compared to other approaches are also reflected in transoral robotic surgery [46,49].

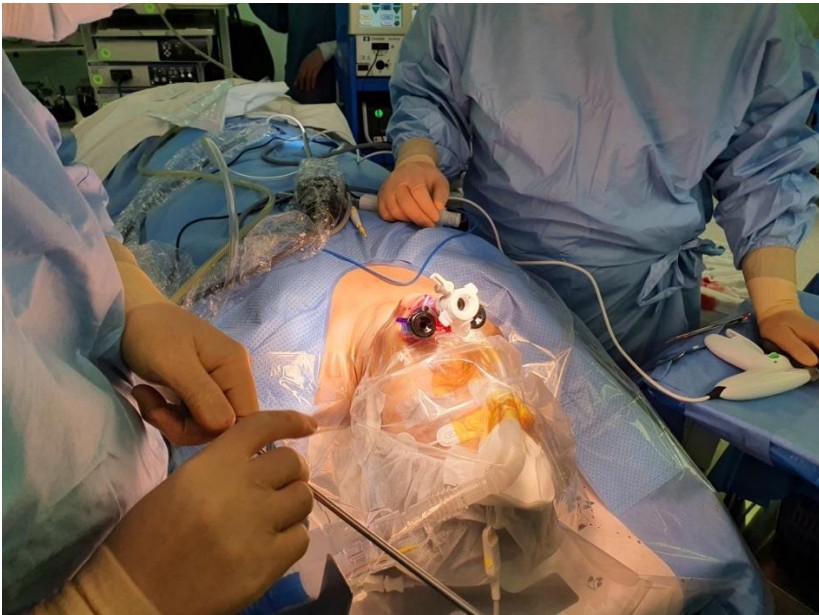

**Figure 3.** Incision sites for transoral vestibule technique.

### 3. Neuromonitoring

*3.1. Prototype of Intra-Operative Neuromonitoring System (IONM)*

Surgeons performing neck surgeries were faced with recurrent laryngeal nerve injuries as there were considerable variation of the course of the nerve, and nerves were often displaced as nearby masses grew larger. Moreover, thyroid operation was especially harder for the right side as the right recurrent laryngeal nerve commonly showed extra-laryngeal branching. To reduce the risk of nerve injuries, Shedd et al., introduced the early version of neuromonitoring 1966 on a canine model [50], then applied it to humans in the same year [51]. The endotracheal balloon was connected to the pressure transduces of a Sanborn pressure recording apparatus, which picked up the electrical response from the recurrent laryngeal nerve stimulation. They have found from this study that there was a distinct pattern of pressure found when the laryngeal nerve was stimulated, which helped them distinguish the presence of nerve injuries. Over the years many different nerve monitoring formats have been introduced, but endotracheal tube-based surface electrodes have become a common monitoring format.

*3.2. Commonly Used Devices*

Current neural monitoring equipment are divided into audio-only systems and both audio and visual systems. Although the mechanisms are similar, the visual system provides exact quantities of amplitudes for each stimulation which helps determine true signals from artifacts. To obtain electrical information, either surface or needle electrodes must be placed onto a patient. Nowadays, needle electrodes are becoming less popular as surface electrodes offer indistinct results to needle electrodes [52,53]. Moreover, needle electrodes may cause trauma, including vocal cord hematoma, laceration, infection, and fracture of the needle. The stimulating electrodes can either be monopolar or bipolar. There have not been studies on which has the better utility, so the usage depends on surgeons' preferences [54].

*3.3. Usefulness of Intra-Operative Neuromonitoring*

Recurrent laryngeal nerve palsy occurs in 1~10% of patients after thyroidectomy [55,56], and injury occurs depending on the level of surgery. Huge tumors with extensive metastasis may obscure vision, or anatomic variations may confuse a surgeon to correctly identify the nerve. Additionally, structural integrity does not necessarily imply that nerve is unharmed. Nerves can be damaged by direct pressure, heat, or stretching. IONM helps for these discreet conditions where nerve appears intact when it is not; then, surgeons performing total thyroidectomy can postpone excision of the contralateral lobe to preserve the other nerve. Many studies have found that IONM to reduce both transient and permanent nerve palsy [57–59].

### 4. Vessel Sealing Devices

As more surgeons adopted vessel sealing devices, the need for individual suture ligation of vessels dramatically decreased. Many research papers have named this technique as 'suture-less thyroidectomy' as these devices replaced traditional clipping, suturing, and tying of vessels. This dramatically simplified the operation steps and thus shortened total operation time [6,7] while showing non-inferiority when preserving the recurrent laryngeal nerve function [9]. The advancements in vessel sealing devices also enabled surgeons to safely perform surgeries such as endoscopic and robotic thyroidectomies, where hand sutures are practically impossible [8]. Currently most commonly used devices in thyroid surgeries are the Harmonic Focus (Ethicon, Cincinnati, OH, USA), Ligasure TM Small Jaw (Covidien, Minneapolis, MN, USA), and Thunderbeat (Olympus, Japan).

### 5. Preservation of Parathyroid Function

Hypoparathyroidism is a common post-operative complication encountered after total thyroidectomy [60–62]. The known risk factors for causing hypoparathyroidism are bilateral thyroid procedures, autoimmune thyroid disease, central neck dissection, substernal

goiter, low-volume thyroid surgeon, malabsorptive state, simultaneous thyroidectomy and parathyroidectomy and prior neck surgery [63]. These risk factors are virtually unavoidable if patients are indicated for such procedures. Therefore, surgeons are required to be extra diligent in preserving parathyroids and their feeding vessels—which is of course, very difficult. That is why recently published meta-analysis revealed median incidence of temporary hypoparathyroidism after thyroidectomy to be 19~38% [64]. To improve outcomes, surgeons currently apply two common techniques preserve parathyroid functions.

### 5.1. Indocyanine Green Angiography (ICG)

Indocyanine green is a water-soluble anionic amphiphilic tricarboocyanine dye that has no side-effects when infused into human bloodstreams. This becomes fluorescent upon excitation by a light with a wavelength in infrared spectrum. Surgeons utilize this technology by monitoring various anatomical structures in real time using a camera specialized for detecting ICG [65]. ICG was implemented in thyroid surgeries to evaluate viability of parathyroids [65–67]. Its efficacy, however, has been questioned due to different grading systems from different authors [66,67]. A recent meta-analysis which analyzed 21 studies have found ICG to be useful in predicting postoperative hypoparathyroidism, but not identifying parathyroid glands during surgery [68]. Possible reasons for its unavailing performance in identifying parathyroid glands might be due to lower resolution of the camera, lack of protocols, and finally the need for surgeons to approach the parathyroid gland before utilizing ICG.

### 5.2. Near Infrared Autofluorescence Imaging

Near infrared (NIR) light sources that have longer excitation wavelengths allows deeper tissue penetration. This property makes NIR light sources adequate for vivo imaging. When fluorescence is injected into the bloodstream, parathyroid glands yield stronger NIR autofluorescence (NIRAF) compared to surrounding tissues. This property aids in surgeons to better identify parathyroid glands during surgery. (1) PTeye and (2) Fluobeam are two early devices that were FDA approved for NIR imaging [69]. Other devices that have been developed are shown in Table 1.

**Table 1.** Specifications of commonly used vessel sealing devices in thyroid surgery.

|                   | Harmonic Focus | Ligasure TM Small Jaw | Thunderbeat |
| ----------------- | -------------- | --------------------- | ----------- |
| Energy used       | Ultrasound     | Bipolar               | Both        |
| Sealing diameter  | 5 mm           | 7 mm                  | 7 mm        |
| Thermal spread    | 3 mm           | 1 mm                  | 3 mm        |

#### 5.2.1. PTeye

PTeye (AiBiomed, Santa Barbara, CA, USA) system consists of a probe that house a NIR light source and an interactive display. When the probe touches a parathyroid gland, it gives a real-time visual and auditory feedback. The initial experience of PTeye by Kiernan et al., in 2021 demonstrated that the device showed similar accuracy rate for identifying a parathyroid gland when compared to an experienced surgeon [70]. However, the difficulty of designing a blinded study makes it hard for proving the added benefit of this new technology.

#### 5.2.2. Fluobeam

Fluobeam (Fluoptics, France) utilizes a handheld camera system that encloses a NIR light source to illuminate the parathyroid gland within the surgical field. The image is processed and visualized on a separate display monitor. Gorobeiko et al., presented a report of 15 cases at utilized this system in 2021 [71]. Their experience with the Fluobeam was almost congruous with the experience of the PTeye, where experienced surgeons easily identified parathyroid glands without the aid of the NIRAF device. They also

noted that unidentified parathyroid glands were not the main cause of post-operative hypoparathyroidism, but rather were due to compromised parathyroid vessels. Examples of commercially available NIRAF devices are listed in Table 2.

**Table 2.** Commercially available Near-infrared autofluorescence devices.

| Name of Device | Producer |
| --- | --- |
| Fluobeam 800 | Fluoptics, Grenoble, France |
| Fluobeam LX | Fluoptics, Grenoble, France |
| PTeye | Medtronic, Dublin, Ireland |
| PINPOINT® + SPY-PHI | Stryker, Kalamazoo, MI, USA |
| Pde-neo II | HAMAMATSU PHOTONICS K.K., Shizuoka, Japan |
| Quest Spectrum® | Quest Meidcal Imaging, Wieringerwerf, The Netherlands |
| EleVision™ IR Platform | Medtronic, Dublin, Ireland |
| IMGE1 S™ RUBINA | Karl Storz, Tuttlingen, Germany |

## 6. Choosing Appropriate Surgery to Minimize Post-Operative Complications

### 6.1. Thyroid Lobectomy vs. Total Thyroidectomy

When patients are diagnosed with differentiated thyroid cancer, radical surgery such as total thyroidectomy with bilateral radical neck dissection will certainly show superior oncologic outcomes compared to a more moderate approach. However, safely removing a tumor without compromising functions and keeping complications to a minimum is very difficult. Balancing the two is still one of the most difficult decisions surgeons must make on every patient. Currently the most widely accepted is the guidelines proposed by the American Thyroid Association in 2015. Following the guidelines, most surgeons would recommend total thyroidectomy when differentiated thyroid cancer patients have gross extra-thyroidal extension, if there are clinical findings of central or lateral neck metastasis, bilateral involvement of multifocal cancers, have history of neck radiation, and if size is large (>4 cm). However, this decision may be difficult for intrathyroidal tumors that are 1–4 cm [72]. A recent meta-analysis on comparison of total thyroidectomy vs. lobectomy on T1-T2N0 differentiated thyroid cancer patients showed non-inferior oncologic outcomes and less complications in lobectomy patients. The overall survival rate of 20-year outcome showed 96.8% vs. 97.4% (OR 1.30, 95% C.I. 0.71–2.37, $p = 0.40$) for total thyroidectomy and lobectomy, respectively. Whereas complications such as bleeding, recurrent laryngeal nerve injury, and permanent hypoparathyroidism were relatively higher in the total thyroidectomy group [73]. Therefore, while total thyroidectomy with radioiodine therapy is justified in patients that possess aggressive traits, lobectomy should be considered for better preserving functions after surgery.

### 6.2. Prophylactic Central Lymph Node Dissection

Many agreements have been made among surgeons and endocrinologists regarding the need for postoperative radio-iodine therapy, and Thyroid Stimulating Hormone (TSH) suppression therapy in thyroid cancers after thyroidectomy. Yet, choosing the extent of lymph node dissection is still an ongoing debate. Prophylactic central neck dissection is defined as complete excision of level VI and VII lymph nodes and currently guideline recommends this procedure when lymph node metastasis is suspected on ultrasonography [72]. However, lymph node metastasis is very common in thyroid cancers, where approximately 35% of the patients are found to have occult central lymph node metastasis upon surgery [74,75]. Thus, routine central lymph node dissection even if there are no visible metastatic lymph nodes might seem like a logical method to secure R0 resection. This routine prophylactic central lymph node dissection (pCND) has been routinely performed mainly in East Asian countries. Yet, in recent years, more data support the disuse of pCND when there is no clinical evidence of metastasis. Randomized control trials have shown that pCND does not change recurrence free survival [75,76]. Moreover, a multicenter study on elderly patients diagnosed with differentiated thyroid cancer have found that pCND may

also increase the risk of post-operative complications, encouraging surgeons to consider a more tailored surgical approach [77].

### 6.3. Risk Stratification and Implementation of Molecular Testing

It is widely accepted that fine needle aspiration (FNA) is a golden standard for diagnosis of thyroid nodules. Yet, nodules that fall into Besthesda III and IV often require surgical excision to decipher whether if they are malignant or not [72]. For such patients, pre-operative risk stratification as well as molecular testing can be used to decide the extent of surgery. Pre-operative risk stratification may involve risk scoring systems devised by the American Thyroid Association (ATA), American Association of Clinical Endocrinologists (AACE), American College of Endocrinology (ACE), and Associazione Medici Endocrinologi (AME) along with findings from ultrasonography. These risk stratification systems are convenient in that they do not require extra laboratory examinations and can be implemented immediately in an outpatient clinic setting. However, not all patients are classified using such scoring systems, and may still require diagnostic thyroidectomy [78,79]. For these patients, molecular tests have become an alternative option in deciding treatment options and preventing unnecessary completion thyroidectomies. Molecular tests involve finding common genetic altercations in thyroid cancers using genetic sequencing methods. Recent studies have demonstrated their positive predictive value to be approximately 97%, and negative predictive value to be 96% [80]. These values are expected to reach higher numbers as new versions of molecular tests, which identifies more mutations, are being developed.

## 7. New Boundaries of Artificial Intelligence in Thyroid Surgery

Artificial intelligence (AI) is a mathematical algorithmic model that replicates cognitive functions of the human brain. This concept was first introduced by McCarthy et al., in the 1950s [81]. However, this field has shown immense potential in the last decade due to the improvement of computer hardware and availability of big data.

### 7.1. Application in Image Recognition

Today, application of AI is implemented mainly in imaging for thyroid diseases. Many studies have pioneered in identification and classification of thyroid nodules using deep learning algorithms [82–84], which has shown much potential in reducing human error in diagnosis of malignant nodules. Most recent meta-analysis compared the sensitivity and specificity of thyroid ultrasound imaging done between AI and radiologists. It has found no significant differences in accuracy for diagnosis of thyroid nodules [85].

### 7.2. Application in Surgery

Fully surgeon-independent surgery is surely the direction we are heading towards. However, it requires leaping more than just few legal issues. A review article by O'Sullivan published in 2019 addressed the issues with applying AI to the surgical field [86]. Other than accountability and liability over legal issues, it has emphasized that an explainable AI is urgently needed in this field. For surgery to be like autonomous driving, the surgeon must be able to realize which direction the AI is going towards with surgery.

## 8. Conclusions

Thyroid surgery have initially begun as bloody and dangerous surgery with high mortality rates. Patients also had to suffer scars on the neck after surgery. As surgeons adapted new technology and novel routes for approaching the thyroid, patients can now enjoy less complications with minimal scars on discrete sites of their bodies. We are now experiencing a new era of surgery. With artificial intelligence adapted smarter and more accurate robotic consoles, we expect another major changes in the field of thyroid surgery in the near future.

**Author Contributions:** Conceptualization, W.K. and J.Y.C.; investigation, W.K. and J.K.L.; resources, W.K., H.W.Y. and J.Y.C.; writing—original draft preparation, W.K.; writing—review and editing, J.K.L., H.W.Y. and J.Y.C.; supervision, J.Y.C.; project administration, J.Y.C.; All authors have read and agreed to the published version of the manuscript.

**Funding:** This research received no external funding.

**Institutional Review Board Statement:** Not applicable.

**Data Availability Statement:** Not applicable.

**Conflicts of Interest:** The authors declare no conflict of interest.

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
