# Peer review of "Advancements in Thyroidectomy: A Mini Review"

_endocrines, doi:10.3390/endocrines3040065_

Round 1

Reviewer 1 Report

In this paper the Authors analayzed the Advancements in Thyroidectomy. It is a debated and novel topic. A comprehensive and extensive literature review of the NCBI database PubMed was also carried out. The article was well conducted and it is interesting in its fields. It is a well-structured paper, written in good English and the References are up dated. 

Minor issues:

In more demolitive surgery, interventions are affected by more severe complications. In the “discussion” section I suggest to better analyze this topic. Therefore, the following paper should be considered:

“Gambardella C, Patrone R, Di Capua F, Offi C, Mauriello C, Clarizia G, Andretta C, Polistena A, Sanguinetti A, Calò P, Docimo G, Avenia N, Conzo G. The role of prophylactic central compartment lymph node dissection in elderly patients with differentiated thyroid cancer: a multicentric study. BMC Surg. 2019 Apr 24;18(Suppl 1):110. doi: 10.1186/s12893-018-0433-0. PMID: 31074400; PMCID: PMC7402571.”

 “Gambardella C, Offi C, Patrone R, Clarizia G, Mauriello C, Tartaglia E, Di Capua F, Di Martino S, Romano RM, Fiore L, Conzo A, Conzo G, Docimo G. Calcitonin negative Medullary Thyroid Carcinoma: a challenging diagnosis or a medical dilemma? BMC Endocr Disord. 2019 May 29;19(Suppl 1):45. doi: 10.1186/s12902-019-0367-2. PMID: 31142313; PMCID: PMC6541563.” 

“Marotta V, Sciammarella C, Chiofalo MG, Gambardella C, Bellevicine C, Grasso M, Conzo G, Docimo G, Botti G, Losito S, Troncone G, De Palma M, Giacomelli L, Pezzullo L, Colao A, Faggiano A. Hashimoto's thyroiditis predicts outcome in intrathyroidal papillary thyroid cancer. Endocr Relat Cancer. 2017 Sep;24(9):485-493. doi: 10.1530/ERC-17-0085. Epub 2017 Jul 10. PMID: 28696209.”

Author Response

Response to Reviewer 1:

Thank you very much for your thoughtful review.

Firstly, I would like to thank you for providing me with great reference articles. I very much agree with your thoughts on radical surgeries having more severe complications. In response to your comments, I have added a new section mentioning the recent paradigm changes in extent of surgery. It reviews how lobectomy is much preferred in certain patients as well as how prophylactic central lymph node dissection for clinically N0 patients should not be recommended as it only increases  post-operative complications.

Reviewer 2 Report

It is a fine and adequate presentation of the evolution of technology in thyroidectomy and the minimal invasive techniques. 

Author Response

Response to Reviewer 2:

Thank you for your thoughtful review.

I have added two more sections providing more insight on recent paradigm shifts in the extent of thyroid surgery and the types of vessel sealing devices endocrine surgeons commonly use.

Reviewer 3 Report

The review appears too superficial and adds nothing new to the literature. Some topics are missing:

vessel sealing devices

different approaches to lymph node dissections

please review the references, as they are too dated

Author Response

Response to Reviewer 3:

Thank you for your thorough review.

Please do understand that as it is a mini-review, I did not go in-depth in many topics but instead tried to provide a brief outline for audiences that may be new to the field of thyroid surgery. In response to topics you have pointed out, I have added a small section reviewing how vessel sealing devices have proven safety and efficacy in recent years. I have also added a section explaining how surgeries are becoming less radical to reduce post-operative complications.

Some references are indeed dated as I used them to reference works that have started new fields: such as early studies of Anuwong et al. on TOETVA.